# A Practical Guide for the Management of Steroid Induced Hyperglycaemia in the Hospital

**DOI:** 10.3390/jcm10102154

**Published:** 2021-05-16

**Authors:** Felix Aberer, Daniel A. Hochfellner, Harald Sourij, Julia K. Mader

**Affiliations:** 1Division of Endocrinology and Diabetology, Medical University of Graz, 8036 Graz, Austria; felix.aberer@medunigraz.at (F.A.); daniel.hochfellner@medunigraz.at (D.A.H.); Julia.mader@medunigraz.at (J.K.M.); 2Interdisciplinary Metabolic Medicine Trials Unit, Medical University of Graz, 8036 Graz, Austria

**Keywords:** steroid induced hyperglycaemia, hospital, practical guide

## Abstract

Glucocorticoids represent frequently recommended and often indispensable immunosuppressant and anti-inflammatory agents prescribed in various medical conditions. Despite their proven efficacy, glucocorticoids bear a wide variety of side effects among which steroid induced hyperglycaemia (SIHG) is among the most important ones. SIHG, potentially causes new-onset hyperglycaemia or exacerbation of glucose control in patients with previously known diabetes. Retrospective data showed that similar to general hyperglycaemia in diabetes, SIHG in the hospital and in outpatient settings detrimentally impacts patient outcomes, including mortality. However, recommendations for treatment targets and guidelines for in-hospital as well as outpatient therapeutic management are lacking, partially due to missing evidence from clinical studies. Still, SIHG caused by various types of glucocorticoids is a common challenge in daily routine and clinical guidance is needed. In this review, we aimed to summarize clinical evidence of SIHG in inpatient care impacting clinical outcome, establishment of diagnosis, diagnostic procedures and therapeutic recommendations.

## 1. Introduction

Steroidal therapies in particular glucocorticoids (GC), represent therapeutic agents of great importance in the treatment and prophylaxis of various acute and chronic inflammatory as well as autoimmune disorders [1]. Despite their efficacy, the use of steroids is associated with a variety of side effects that can be pragmatically divided into three categories: (1) Immediate side effects include the occurrence of fluid retention with oedema, blurriness of vision, impairments of mood, immune response modulation and the development of steroid induced hyperglycaemia (SIHG). (2) Idiosyncratic side effects summarize the development of avascular necrosis, cataract formation, glaucoma and psychosis. (3) More gradual side effects affecting the endocrine system and inducing the development of bone disease, dyslipidaemia, obesity and adrenal suppression [2]. As GCs decrease peripheral insulin sensitivity, increase hepatic gluconeogenesis, trigger insulin resistance on the level of the lipid metabolism and adipose tissue, as well as inhibit pancreatic insulin production and secretion, they represent a drug class with the highest risk of provoking the development of hyperglycaemia and overt diabetes mellitus (DM) [3,4,5].

## 2. Prevalence of Steroid Induced Hyperglycaemia in the Hospital

The prevalence of SIHG is dependent on the dose, indication and setting of use. Individual conditions such as age, baseline body mass index (BMI) and family history of diabetes are known to impact the risk of SIHG development. Older observational data indicate that 2% of incident diabetes cases in a primary care population are associated with GC therapy and the odds ratios for presenting with new-onset diabetes after introduction of GCs in various studies has been described to range from 1.36–2.31 [6]. Patients without history of diabetes developed in-hospital hyperglycaemia (≥10 mmol/L; ≥180 mg/dL) in 70% when relevant doses of GCs were administered [7]. A meta-analysis summarized studies in which patients without pre-existing diabetes who received systemic GCs and showed a rate of SIHG development in 32.3% and in further 18.6% diabetes was sustainable during the follow up [8]. In patients who received solid organ transplantation and GC therapy the prevalence was described to be between 17% and 32% [9,10] and in a high-risk population of people who received high dose systemic therapy for acute Graft-versus-Host disease, two thirds of the cohort showed median glucose readings in the hyperglycaemic range (defined as fasting glucose ≥7 mmol/L or ≥126 mg/dL) [11]. As SIHG is suggested to be a transient problem resolving after the discontinuation of GCs, data indicate that diabetes can persist and GCs just unmasked a pre-existing glucose metabolism disorder [12,13]. While mainly systemic steroids were identified to expose the patient to an increased risk for hyperglycaemia, recently, topically used GCs were also shown to be associated with an elevated risk of diabetes [14].

## 3. Impact of Steroid Induced Hyperglycaemia

It has been shown that acute and chronic hyperglycaemia that are present in many cases in the hospital setting are important risk factors for prolonged hospital stays, infectious complications, poorer surgical outcomes and increased mortality [15,16,17]. Data of people with SIHG without previously known DM are scarce. Some studies reported an association of reduced response to chemotherapeutics and increased mortality in patients with haematological disorders as well as in patients with solid cancer [11,18,19,20,21]. Similar outcome with inferior prognosis is reported for patients who underwent kidney transplantation [22] or in case of being hospitalized for acute exacerbated chronic obstructive pulmonary disease [23]. However, the question whether elevated blood glucose is just a surrogate parameter for severe illness and adverse outcome [24] or if blood glucose might be a modifiable risk factor has not been answered yet. Given the fact that most of the studies yet performed employed observational designs and data derived from randomized controlled clinical studies are still substantially lacking, uncertainties whether asymptomatic and transient inpatient hyperglycaemia should be treated, remain [25].

## 4. Definition of Steroid Induced Hyperglycaemia

SIHG is defined as abnormally elevated blood glucose associated with the use of GCs in patients with or without pre-existing DM. The diagnostic criteria for SIHG do not differ from other types of diabetes and include a confirmed fasting blood glucose ≥7 mmol/L (≥126 mg/dL), a glucose level of ≥11.1 mmol/L (≥200 mg/dL) at 2 h following ingestion of 75 g glucose in an oral glucose tolerance test (OGTT), an HbA1c ≥6.5% (≥48 mmol/mol) or a random blood glucose ≥11.1 mmol/L (≥200 mg/dL) [26]. However, in patients with SIHG, diagnosis can be more challenging: fasting blood glucose might be normal especially when short- or intermediate-acting GCs are administered in single morning doses. Apart from its difficulties in implementation of oGTT in hospitalized patients, hyperglycaemia might be absent after glucose exposure in an oGTT, especially when it is performed in the morning when the diabetogenic effect of the GCs is not yet present. HbA1c might be inconspicuous especially in those with new-onset GC therapy as it reflects the glycaemic situation in the weeks prior to the time point of measurement. In addition, several conditions such as chronic kidney disease or hemoglobinopathies, that are frequently present in people requiring steroids, affect the reliability of HbA1c measurements. Nevertheless, determination of HbA1c can be useful to evaluate glycaemic control in patients who are on long-term GC therapy or to distinguish between new-onset diabetes and pre-existing DM in a situation of hyperglycaemia after GC initiation. Due to the mentioned limitations of the usual diagnostic approach to detect SIHG, it is recommended to perform frequent (capillary) glucose monitoring in those who receive high doses of GCs (defined as >20 mg prednisolone or equivalent). This approach is particularly recommended in people with a high risk to develop SIHG (e.g., advanced age, higher BMI, previously present impaired glucose tolerance, prediabetes or family history of diabetes). Then, a random glucose value ≥11.1 mmol/L (≥200 mg/dL) can be utilized to establish the diagnosis of SIHG [27].

## 5. Treatment Targets

No clear evidence is available for the establishment of therapeutic goals for patients with SIHG [28]. According to the American Diabetes Association (ADA) glucose targets for patients with SIHG do not differ from those with any other type of diabetes and should be individualized according to specific factors such as life expectancy, comorbidities, patient compliance and risk of hypoglycaemia [29]. In hospitalized patients a target glucose range of 7.8–10.0 mmol/L (140–180 mg/dL) is recommended for the majority of critically and non-critically ill patients. More stringent goals such as 6.1–7.8 mmol/L (110–140 mg/dL) may be appropriate for selected patients, if this goal can be achieved without relevant hypoglycaemia [30]. However, when aiming to achieve lower target glucose levels it has to be considered that people with SIHG often suffer from severe underlying disease (e.g., cancer), are in the perioperative care setting (e.g., recently transplanted patients or those requiring steroids as supportive therapy [31]), receive concomitant complex therapies (chemotherapy, immunosuppressants, antimicrobial therapy, etc.) and thus, might be prone to larger glucose fluctuations. In the course of treatment, GCs need frequent dose adaptions that result in altered requirements of glucose lowering therapies. As a consequence, the risk for hypoglycaemia is increased when stringent glucose targets were chosen. Therefore, in specific patient populations (incurable disease with short life expectancy, advanced age and comorbidities, susceptibility for hypoglycaemia and impaired awareness of hypoglycaemia) the major aim will be the avoidance of hypoglycaemia and hyperglycaemic symptoms [29,32,33].

## 6. Admission to the Hospital

HbA1c should be assessed in all people with previous DM in order to evaluate glycaemic control prior to GC initiation. In people who were not previously diagnosed with DM and who require relevant amounts of GCs (>20 mg prednisolone or equivalent) or who are at high risk to develop diabetes or SIHG (criteria see Figure 2), HbA1c should be assessed at admission [34]. This helps to distinguish whether a pre-existing unrecognized DM is present which would result in a more pronounced glycaemic excursion following GC therapy initiation. It can also be assumed that hyperglycaemia is self-limiting after cessation of GC treatment and glucose levels return to normal, given that HbA1c levels were inconspicuous prior to GC treatment [35]. A position statement released by the Joint British Diabetes Societies (JBDS) has defined an algorithm for glucose monitoring in hospitalized patients requiring GC treatment. They postulate that determinations of glucose should be performed at least once daily, preferably prior to lunch or 1–2 h post lunch or before the evening meal in people without diabetes in whom GC therapy was initiated. Once daily glucose measurements should be continued and if glucose readings exceed 11.1 mmol/L (200 mg/dL) repeatedly, frequency of testing should be increased to 4 times daily (before each meal and at bedtime) which is mandatory in patients with pre-existing diabetes treated with GC [33]. In the course of hospitalization and scheduling of GC initiation, patients should be well informed about potential side effects of GCs including SIHG and its therapeutic consequences.

## 7. Initiation of Glucose Lowering Therapy

A practical approach when the implementation of glucose lowering therapy should be initiated was published by Suh et al. who recommend initiating therapy when pre- or post-prandial glucose repeatedly exceed 7.8 (140 mg/dL) or 11.1 mmol/L (200 mg/dL), respectively [28,33]. Similar to the management strategies to lower glucose in patients with type 2 diabetes (T2DM), stepwise intensification of antihyperglycaemic therapy and frequent re-evaluation should be performed in SIHG. The glucose lowering agents of choice should match daily glucose profiles and the mechanism of action should fit to the corresponding GC agent.

## 8. Treatment of Steroid Induced Hyperglycaemia in the Hospital

### 8.1. Oral Antihyperglycaemic Agents

In the outpatient setting some oral hypoglycaemic agents (OHA) might have the potential to improve glycaemic control and prevent or delay the development of SIHG [36,37]. There is only very little evidence available showing clinical efficacy of using OHA for in-hospital hyperglycaemia caused by GCs. Insulin sensitizers such as metformin and pioglitazone might be used to enhance insulin sensitivity and reduce insulin resistance [38,39,40,41] and can be continued in preexisting T2DM unless contraindications exist. However, in hospitalized patients, specifically in those who are acutely ill, susceptibility to hypoxia or acute kidney injury as well as fluid retention can limit the use of these agents. In addition, in particular pioglitazone, needs an expanded time to exert full action which disqualifies it to be applied for acute SIHG. Insulin secretagogues, stimulating endogenous insulin production might be suitable to tackle mild SIHG in the inpatient setting, specifically in inpatients who are non-severely ill and who receive short-acting steroids once daily in the morning [33]. However, insulin secretagogues should be used with caution as there is an increased risk of hypoglycaemia especially when steroid doses are tapered or meals are skipped. The safe side effect profile of incretin mimetics such as DPP4 inhibitors might support their application in hospitalized patients with SIHG, their acute glucose lowering effect is of moderate extent and mostly they are mostly used as an adjunct to insulin therapy. The use of GLP1-receptor agonists bears the risk of gastrointestinal adverse effects in particular during the initiation phase which limits their broad usage for acutely ill, hospitalized patients with SIHG [28]. The use of the sodium-glucose co transporter-2 (SGLT2) inhibitor dapagliflozin has shown to be safe in patients hospitalized for chronic obstructive pulmonary disease (COPD) developing SIHG, but did not improve glycaemic control or clinical outcomes [42].

OHAs might be an adequate choice in inpatients with stable and non-critical disease and mild hyperglycaemic excursions. In those with significant hyperglycaemia and severe illness, insulin remains the treatment of choice in the hospital setting as also suggested by the current guidelines for inpatient diabetes management [30].

### 8.2. GC Dependent Glucose Increase and the Choice of Insulin Therapy

The hyperglycaemic effect of different GCs can be pragmatically transferred to the pharmacokinetic profiles of different GCs. Thus, the insulin therapy chosen for SIHG has to take the used agent, the current dose, the time point and interval of the GC administration into account. Table 1 summarizes the pharmacokinetics of available GCs adapted from the literature [43,44], Table 2 indicates potential glucose profiles according to the administered GC agent.

The upcoming paragraphs describe different scenarios of patients with normal glucose homeostasis under regular conditions as well as patients with T2DM well-controlled under dietary recommendations or treated with OHA in whom relevant hyperglycaemia is a consequence of GC administration who subsequently require insulin therapy. The paragraphs contain recently available recommendations which were given for people with new-onset hyperglycaemia or previously known T2DM.

#### 8.2.1. Scenario 1: Short-Acting Glucocorticoids (Hydrocortisone)

Short-acting hydrocortisone has a considerably high mineralocorticoid activity and is therefore suitable as first-line agent in the therapy of adrenal insufficiency. In its usual application as hormone replacement therapy, hydrocortisone should not cause relevant hyperglycaemia if the substance is administered in physiological doses. For these reasons, no data of SIHG induced by short-acting GCs are available and the recommendations arise from speculations. However, the required physiological doses are often overestimated and exogenous Cushing syndrome including SIHG can occur [45]. In addition, in specific conditions such as acute illness, stress or during surgery substantial dose increases can be required that might induce SIHG. Hydrocortisone is characterized by a fast onset and short duration of the intended effect. Simultaneously, the expectable glucose profile in selected patients will show to have a fast and strong increase but only of short duration. Hence, these commonly transient and mostly self-limiting glucose peaks remain often unrecognized. Whether these short-term hyperglycaemic episodes require glucose lowering therapy has to be decided on an individual basis. In patients with significant hyperglycaemia or impaired health status, the agent of choice is short-acting insulin (rapid-acting insulin analogues or regular insulin) which should be injected at the time or shortly after GC administration. As hydrocortisone is usually administered twice or thrice daily, multiple rapid-acting insulin doses might be suitable to improve glycaemic control. However, it has to be taken into account that morning doses during replacement therapy are usually higher than doses throughout the day and insulin requirements thus might be lowered subsequently. Initiation of the dose can be recommended with 0.1 IU/kilogram (kg) bodyweight (BW) [46]. In addition, insulin therapy can be intensified by including insulin corrections in case of higher subsequent glucose values or persisting post-prandial hyperglycaemia assuming that the intensification requires pre/post-prandial glucose assessments. In these cases, schematic increments of 0.04 IU/kg for pre-prandial values from 11.1–16.7 mmol/L (200–300 mg/dL) or 0.08 IU/kg for values ≥ 16.7 mmol/L (≥300 mg/dL) can be added to the scheduled insulin dose. It is important to mention that insulin requirements are GC dose-dependent; hence, reduction of GC is usually related to an improvement of glycaemia. Reduction of rapid-acting insulin should be performed proportionally to the reduction in GC dose, vice versa rapid-acting insulin dose can be increased when doses of GCs are recommended to be increased [2,46].

#### 8.2.2. Scenario 2: Intermediate-Acting Glucocorticoids (Predniso(lo)ne and Methylprednisolone)

Intermediate-acting glucocorticoids represent the most commonly prescribed steroid agents. Their high glucocorticoid activity makes them useful for long-term anti-inflammatory and immunosuppressant treatment especially in solid-organ transplant patients and those with COPD. Considering a single dose administration in the morning, which corresponds to the typical prescription, hyperglycaemia develops slowly, but continuously, mostly lasts until the evening and gradually recovers until the next morning simultaneously following the peak and duration of action of the steroid agent. To best fit this glucose pattern short- or intermediate-acting basal insulins such as insulin detemir or NPH (neutral protamine Hagedorn) insulin is recommended. A clinical recommendation to initiate insulin was issued by Clore et al. who suggest initiating a weight-dependent scheme with 0.4 IU/kg of NPH insulin [47]. Another study described clinical efficacy when lower doses of NPH (0.2–0.3 IU/kg) dependent of the GC dose were administered and whether patients were fasting or not [48]. While the kinetics of intermediate-acting glucocorticoids appear to fit best to the glucose lowering property of NPH insulin, two randomized studies with insulin glargine U100 at a fixed starting dose of 0.5 IU/kg [49] or initiated according to admission glucose (0.3 or 0.4 IU/kg) [50] demonstrated non-inferiority compared to NPH insulin in regards of efficacy and safety, including nocturnal hypoglycaemia. A sufficient performance was also confirmed in a study which used insulin glargine U100 incorporated in a clinical decision support system for the treatment of in hospital SIHG [51]. Probably a reasonable and simple approach is to initiate basal insulin in a GC dose-dependent dose, starting with 0.1 IU/kg BW if patients receive 10 mg of prednisone or equivalent and 0.2 IU/kg BW in case GC dose is 20 mg, 0.3 IU/kg BW when dose was set at 30 mg and so on [47,52]. Insulin dose finding based on patient age and kidney function has been proposed, indicating that initial doses should be lower in those with impaired kidney function (eGFR < 30 mL/min/1.73 m^2^) or older than 70 years [17,33]. Subsequent dose adjustments should be based on achievement of glycaemic targets assessed by glucose measurements performed the next morning given that the GC is taken in the morning. Multiple daily administrations of intermediate-acting GCs are more complex as hyperglycaemia might overlap and persistent hyperglycaemia can occur (see glucose profile in Table 2). In this case, NPH insulin once daily will not be sufficient and NPH twice daily or a switch to longer-acting insulin (e.g., glargine) is required. If necessary, additional rapid-acting insulin boluses might be added. This can be established by either correctional bolus insulin (correction factor see scenario 1) or by switching to premixed insulin with a mixture of 70% rapid-acting and 30% basal insulin administered simulously to the GC intake [46].

#### 8.2.3. Scenario 3: Long-Acting Glucocorticoids (Dexamethasone)

Dexamethasone, as the most potent GC agent, is characterized by a prolonged duration of action lasting for more than 24 h. It is clinically used in various scenarios such as in inflammatory diseases, as an analgesic or for the reduction of brain pressure in cerebral cancer or cerebral edema. In the recent severe acute respiratory syndrome coronavirus type 2 (COVID-19) pandemic, dexamethasone has been recommended for those with impairments in gas exchange due to viral pneumonia [53], irrespective of diabetes status. This approach needs to be further investigated in people with diabetes as deterioration of glycaemic control and new-onset hyperglycaemia were associated with inferior outcome in people with COVID-19 [54,55,56,57]. For hyperglycaemia during dexamethasone treatment for COVID-19, Rayman et al. have recently published a guidance article. In insulin naïve patients, they recommend to start NPH insulin when glucose exceeds a threshold of 12 mmol/L (~216 mg/dL) in a dose of 0.3 IU/kg/day while 2/3 should be administered in the morning and the remaining third in the evening. They also propose a dose reduction to 0.15 IU/kg in case of age >70 years or eGFR below 30 mL/min. Insulin doses are recommended to be titrated according to morning or evening glucose vales in a manner of a reduction of 20% if glucose falls below 4.1 mmol/L (~70 mg/dL) or decreased by 10% in case of glucose between 4.1–6.0 mmol/L (~70–110 mg/dL). Vice versa, insulin dose should be up-titrated by 20% if glucose values exceed 18 mmol/L (~320 mg/dL) and by 10% if glucose values are between 12.1 and 18 mmol/L (~220–320 mg/dL) [58]. In general, hyperglycaemia in association with long-acting GCs, which are usually administered in the morning, develops slowly, peaks during the day (varying time point) and is sustained for 24 h after intake. Thus, intermediate-acting basal insulins (NPH insulin, insulin detemir) should be prescribed twice daily (initial dose 0.3 IU/kg BW). Alternatively, long- or ultralong-acting basal insulin analogues (insulin glargine U100/U300 or insulin degludec) might be the most appropriate insulin to control hyperglycaemia in this situation (initial dose 0.2 IU/kg BW). Insulin dose should be adjusted according to glucose 24 h after GC intake and onset of hyperglycaemia. To date, to the best of our knowledge, not a single study has been conducted to test new generation ultra-long-acting basal insulin analogues for the treatment of SIHG.

### 8.3. Insulin Intensification and Adjustments

Especially in those without pre-existing diabetes prior to GC treatment, it is of utmost importance for insulin titration to know current GC dose and GC dose changes (tapering or increase). In a pragmatic approach, insulin dose can be adjusted by half the percentage of the GC dose change. For example, when GCs are increased or tapered by 50%, insulin dose is suggested to be increased or reduced by 25%, respectively. In patients with pre-existing DM a deterioration of glycaemic control secondary to GC therapy can be expected. In this regard, type of GC agent as well as time point and interval of GC application have to be taken into account.

#### 8.3.1. Adjustment of Basal Insulin Therapy

When basal insulin therapy was already initiated, up-titration by 10–20% should be performed in case of sustained hyperglycaemia (fasting glucose exceeding 11.1 mmol/L [200 mg/dL]) on 2–3 subsequent days [17,33]. Alternatively, adjustments can be performed in 2 IU increments (conservative approach) to reach the individual glucose target; however, a steady dose adjustment must be warranted. Persisting hyperglycaemia despite basal insulin titration with predominantly postprandial hyperglycaemia requires additional rapid-acting insulin administrations either as rapid-acting insulin injection or incorporated in premixed insulins.

#### 8.3.2. Adjustment of Rapid-Acting Insulin Therapy

Rapid-acting insulins should be primarily administered at the time point of GC administration and can be initiated with 0.1 IU/kg BW. In addition, rapid-acting insulin should be used to correct pre-prandial and spontaneous hyperglycaemia. In such cases add-on of 0.04 IU/kg for pre-prandial values from 11.1–16.7 mmol/L (200–300 mg/dL) or 0.08 IU/kg for values ≥ 16.7 mmol/L (≥300 mg/dL) can be additionally added to the scheduled insulin dose. It is important to mention that insulin requirements depend on GC dose; hence, reduction of GC is usually accompanied by an improvement of glycaemia. Reduction of rapid-acting insulin should be performed proportionally to the reduction in GC dose, vice versa rapid-acting insulin dose can be increased when GC doses are increased [2,46].

#### 8.3.3. Adjustment of Basal-Bolus Insulin

In patients with pre-existing basal-bolus insulin therapy doses of basal and bolus insulin should be adjusted according to the above recommendations. However, those with endogenous insulin deficiency (as people with type 1 diabetes) are more prone to hypoglycaemia which has to be considered when doses are increased [59]. A specific approach how to adjust insulin in people with preexisting type 1 diabetes is given in Section 8.3.5.

A schematic algorithm for the initiation and intensification of glucose lowering therapy in SIHG is illustrated in Figure 1. This algorithm is not valid for patients with preexisting type 1 diabetes.

#### 8.3.4. Adjustment of Insulin Therapy in Patients with Type 1 Diabetes (T1DM) 

It has been shown that relevant doses of transiently administered GCs (in the referenced study 60 mg prednisone/day) lead to an increase in insulin requirements of 70% on average with considerable inter-individual variation to normalize blood glucose levels in patients with previously known well controlled T1DM. This glucose increase was sustained the day after GC therapy was discontinued, indicating a longer lasting hyperglycaemic effect despite the use of an intermediate-acting GC agent. Interestingly, the GC induced additional insulin requirements to achieve reasonable glycaemic control varied considerably and independently from previous insulin dose (30–100% increase) which makes recommendations for adjustments challenging. [60]. Dashora et al. described a 50% increase in insulin requirements in females with T1DM requiring variable doses of GC therapy for treatment of hyperemesis gravidarum [61].

Due to the heterogeneity of the effect of GC on glucose metabolism in patients with T1DM it is recommended to intensify frequent monitoring of glucose upon initiation of GC therapy, as deterioration of glycaemic control has to be expected. As patients with T1DM are more prone to hypoglycaemia in comparison to patients with T2DM, initial dose adjustments have to be taken very carefully and in an iterative manner [33]. GCs are a well-known trigger for diabetic ketoacidosis in patients with deficiency or absence of endogenous insulin secretion, thus proper insulin dose adjustments to the GC therapy are recommended and transient hyperglycaemia should not be trivialized in these patients. Clinical evidence for insulin dose adjustments for patients with T1DM on GC therapy both, in the inpatient or in the outpatient setting, is largely lacking and not described in detail in any treatment guideline [62]. Moreover, the present article discusses SIHG in the hospital setting where besides GC therapy, numerous other factors such as acute disease and altered daily routine additionally influence glucose control. As there is only very little evidence available we suggest a cautious increase in total daily insulin dose (TDD) according to prednisolone (or prednisolone equivalent [PE]) dose, a suggestion that needs further scrutiny in clinical practice:PE of 20 mg → 10% increase in TDDPE of 40 mg → 20% increase in TDDPE of 60 mg → 30% increase in TDD

Taking these estimations into account the following considerations are important:Adjustments of insulin therapy when short-acting steroids (hydrocortisone) are used:○If short acting GCs are used, then an increase of rapid-acting insulin at the time point of GC intake might be sufficient. A correctional rapid-acting insulin dose can be administered in case of persistent hyperglycaemia after 3–4 h when the rapid-acting insulin action has tapered off. As a consequence, the ratio of rapid-acting to basal insulin will exceed the usual 50:50 ratio.Adjustments of insulin therapy when intermediate-acting steroids (e.g., prednisolone) are used:
○Approach A: An increased dose of rapid-acting insulin at the time of intermediate-acting prednisolone administration might be appropriate aiming to achieve glucose control at noon.○Approach B: In case of pre-existing therapy with intermediate-acting basal-insulins (NPH insulin or insulin detemir) that are usually injected twice daily, a dose increases at the time point of GC intake (usually in the morning) is recommended.○Approach C: In patients previously using (ultra-)long acting basal-insulins (insulin glargine U100/U300 or insulin degludec), approach A might be sufficient; in case of an expected long-term GC treatment, these patients might benefit most from a switch to intermediate-acting basal insulins (NPH insulin, insulin detemir. In such case, the basal insulin should be injected twice daily with a proportionally higher dose at the time point when the GC agent is administered.Adjustments of insulin therapy when long-acting steroids (e.g., dexamethasone) are used:○Long-acting GCs will trigger continuous and long-lasting hyperglycaemia over 24 h, thus it might be suitable to adjust the total daily basal-insulin dose according to the GC dose as outlined above.

Of note, the continuation of using preexisting insulin pump therapy (continuous subcutaneous insulin infusion [CSII]) in the hospital is not recommended in the majority of cases especially in those who are acutely hospitalized and severely ill. In patients without physical or mental disorders, the self-managed continuation of CSII therapy might be justified [63,64]. CSII systems provide adjustable basal rates, programming of different basal rate profiles as well as a temporary % increase/decrease of the current basal rate. Moreover, bolus dosing can be performed more frequently to administer correctional insulin when required without an additional injection as in pen-based therapy. Thus, in insulin pump users the continuation of insulin pump therapy with according to adjustment of insulin dose might be a considerable option if deemed practicable by the physicians in charge. However, clinical evidence supporting this presumption is not available yet.

In summary, the adjustment of insulin doses in patients with complex previous insulin therapies (i.e. T1DM) can be performed according to the above recommendations, which are quite carefully elaborated, but still require additional individualization, frequent glucose monitoring and close-meshed therapy adjustments.

#### 8.3.5. The Critically Ill Patient

Hyperglycaemic derailments as well as severe and subsequent hypoglycaemia might complicate the clinical course in patients hospitalized on intensive medical care units and might impact on adverse outcomes. During critical illness, factors such as stress, inflammation, failure of kidney function or administered therapeutics, specifically GCs, detrimentally impact on glucose metabolism in people with and without previously known diabetes. In most of patients with critical illness and hyperglycaemia, insulin therapy should be introduced as continuous intravenous application [30]. Intravenous insulin provides the advantage of more rapid insulin adjustments to hyperglycaemic levels. Rapid-acting human insulin or analogues should be prepared by 50 IU rapid-acting insulin mixed with 50 mL sodium chloride (0.9%) with a starting dose of 0.1 IU/kg/h [64]. The switch to subcutaneous insulin is recommended when patient status improves (e.g., uptake of oral nutrition, scheduled transfer to general ward) and metabolic status is balanced. Basal insulin can be started at a dose of 50% of the previous 24 h insulin dose as administered intravenously in an overlapping manner (basal insulin application 2 h prior to cessation of intravenous insulin) in order to prevent rebound hyperglycaemia or acidosis [64]. Of course, also critically ill patients with SIHG should be treated with intravenous insulin, however, no specific recommendations for the treatment with intravenous insulin differing from the recommendation in “usual” critical care hyperglycaemia are available. The used insulin dose of the intravenous insulin application can help to estimate the appropriate dose of subcutaneously administered insulin.

## 9. Discharge from the Hospital

GCs frequently need to be continued after the inpatient stay and hence, hyperglycaemia also might persist [12]. Of note, hyperglycaemic state remains often also in those where GC therapy was discontinued indicating that people with SIHG are prone to develop T2DM.

Based on recommendations of a guideline published by the Joint British Diabetes Societies [33], it is necessary that all patients should be informed about the nature of SIHG, symptoms of hypo- and hyperglycaemia and its consequences if not properly treated. If applicable, patients should be trained in the use of insulin pens and self-monitoring of blood glucose (SMBG). Patients should be advised in the frequency of necessary SMBG and recommended to document glucose values and if applicable insulin doses. Optimally, all patients, irrespective of diabetes therapy at hospital discharge should have access to adequate glucose monitoring technology at home in order to avoid subsequent relevant hyperglycaemia. Adequate and individualized treatment plans should be made available to patients to avoid consecutive presentations at emergency departments potentially resulting in hospital readmissions. Individual therapy regimens should be prepared which contain recommendations for insulin dosing and which give a chance of self-adjustments especially taking into account possible dose changes in the GC therapy. Patients should be offered the possibility to regularly contact the medical staff of the outpatient clinic in case of concerns or problems regarding current glycaemic control. The general practitioner should be introduced in the case and preferentially take the lead concerning the management of the hyperglycaemic state. In addition, HbA1c should be measured every three months [65]. A possible admission and discharge algorithm is illustrated in Figure 2.

## 10. Discussion

GCs are frequently prescribed as they have been confirmed to potentially improve outcomes in various autoimmune and inflammatory diseases, as well as recently also in COVID 19 [66]. Relevant doses of GC therapy potentially lead to hyperglycaemia in both hospitalized patients and patients in outpatient care exposing them to a higher risk of acute and chronic complications.

Diagnosis, monitoring and in particular the management of SIHG represents an everyday challenge and often physicians not specifically working in the field of diabetes and endocrinology have to deal with the management. Therefore, this review aims to provide summary figures that can be used by clinicians for the management of SIHG in routine care. There is a large amount of data available which confirmed the potential burden of chronic hyperglycaemia in both type 1 and type 2 diabetes. In contrast, whether mild, asymptomatic and mostly transient hyperglycaemia, specifically in hospitalized patients impacts on outcome, has not been systematically investigated, specifically not in those with SIHG [25]. Observational data identified elevated glucose as potential biomarker for adverse surgical outcomes, longer hospital stays and an increased mortality [67,68,69]. Vice versa, aggressive glucose lowering therapy in hospitalized patients, in particular using insulin with the risk of causing hypoglycaemia has also shown to negatively impact mortality [70]. For SIHG, which represents a different but important entity, clinical data, both from observational and interventional trials, is largely lacking.

Certainly, there is an increasing need to provide more evidence which, first, identifies the most safe and efficient therapy modalities to treat SIHG, secondly, sets a basis for defining recommended glucose targets and thirdly, allows to answer the question whether improved glycaemia translates to superior outcomes. In addition, novel diabetes technology such as continuous glucose monitors, electronic decision support systems and automated insulin delivery systems might be beneficial to better control SIHG and provide more outcome data in the near future.

## Figures and Tables

**Figure 1 jcm-10-02154-f001:**
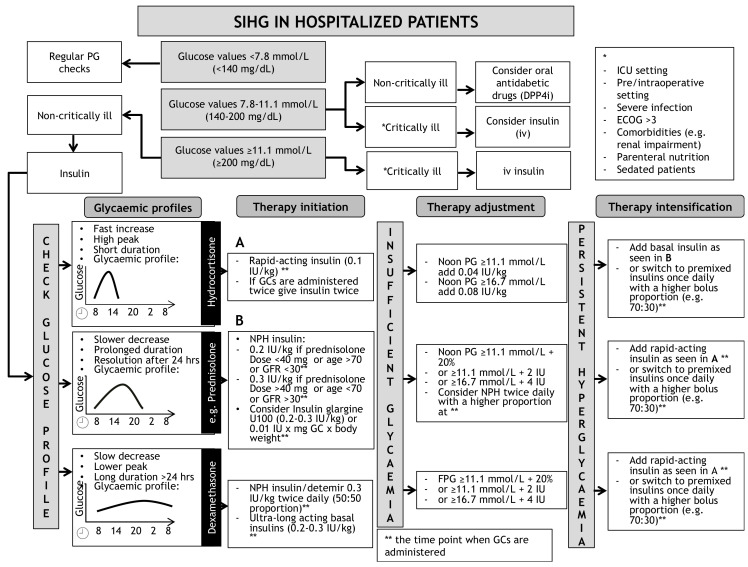
Opinion-based schematic algorithm for initiation, adjustment and intensification of insulin therapy for treatment of SIHG. DPP4i = Dipeptidyl-Peptidase4-inhibitor, ECOG = Karnofsky index, FPG = Fasting plasma glucose, GC = Glucocorticoid, ICU = Intensive Care Unit, IU = International Units, NPH = Neutral Protamine Hagedorn, SIHG = Steroid induced hyperglycaemia. * = definition of critical illness, ** = indicating the time point when glucocorticoids are administered. (**A**) indicates recommendations for initiation of rapid-acting insulin. (**B**) indicates recommendations to initiate basal in-sulin.

**Figure 2 jcm-10-02154-f002:**
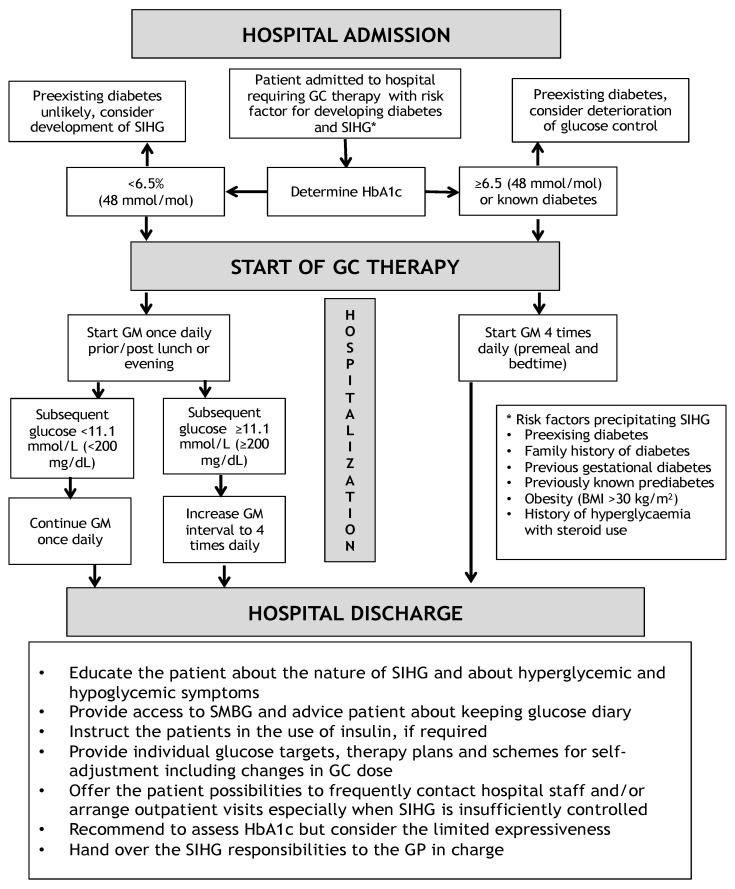
Opinion based admission and discharge algorithm for hospitalized patients with SIHG modified from [33]. BMI = Body mass index, GC = Glucocorticoids, GM = Glucose Monitoring, GP = General practitioner, SIHG = Steroid induced hyperglycaemia, SMBG = self-monitored blood glucose. * = risk factors for steroid induced hyperglycaemia.

**Table 1 jcm-10-02154-t001:** Different corticosteroids and their equivalent doses, steroidal kinetics and potential to trigger hyperglycaemia.

Glucocorticoids		Approximate Equivalent Dose (mg)	Plasma Peak Concentration (minutes)	Elimination Half-Life (hours)	Duration of Action (hours)	Hyperglycaemic Effects (hours)		
						Onset	Peak	Resolution
Short-acting	Hydrocortisone	20	10	2	8–12	1	3	6
Intermediate-acting	Predniso(lo)ne	5	60–180	2.5	12–36	4	8	12–16
	Methylprednisolone	4	60	2.5	12–36	4	8	12–16
Long-acting	Dexamethasone	0.75	60–120	4	36–72	8	variable	24–36

**Table 2 jcm-10-02154-t002:** Schematic illustration of different glucocorticoids and their potential effect on glycaemia. Long-acting agents are usually administered only once daily. These examples are presuming people with normal glucose homeostasis prior to start of glucocorticoid therapy. X-axis: time of the day; y-axis: potential influence on glucose.

Glucocorticoids		Hyperglycaemic Effects (hours)			Glucose Profiles (GC Given Once Daily [8 a.m.])	Glucose Profiles (GC Given Twice Daily [8 a.m. and 20 p.m.])
		Onset	Peak	Resolution		
Short-acting	Hydrocortisone	1	3	6	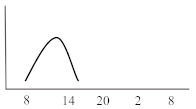	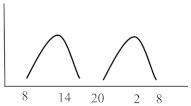
Intermediate-acting	Predniso(lo)ne	4	8	12–16	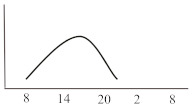	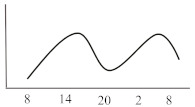
	Methylprednisolone	4	8	12–16		
Long-acting	Dexamethasone	8	variable	24–36	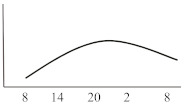	n.a.

## Data Availability

Not applicable.

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
