# Peer review of "A Practical Guide for the Management of Steroid Induced Hyperglycaemia in the Hospital"

_jcm, 2021, doi:10.3390/jcm10102154_

Round 1

Reviewer 1 Report

When given glucose values in mmol/L include the equivalent value in mg/dL in parenthesis.

Would include references on the use of insulin drips for SIHG in critically ill pts and also as a way to calculate total daily insulin requirements in some hospitalized patients.

Would also include references regarding use of insulin analogs plus NPH insulin for treatment of SIHG.  

Author Response

We thank the reviewer for taking his time to read our manuscript and the suggestions given.

When given glucose values in mmol/L include the equivalent value in mg/dL in parenthesis.

In the revised manuscript, we also provide glucose unit equivalents in mg/dl. 

Would include references on the use of insulin drips for SIHG in critically ill pts and also as a way to calculate total daily insulin requirements in some hospitalized patients.

Thank you for this suggestion. We are aware that SIGH is also present in critically ill patients. We have added a separate paragraph describing the critically ill patient.

Would also include references regarding use of insulin analogs plus NPH insulin for treatment of SIHG.

In this matter, we refer to line 229 where we considered the use of rapid acting human insulins analogues as equivalent. The available evidence does not allow to prefer one of these in the matter of SIHG treatment. The use of NPH insulin plus rapid-acting insulin is described in figure 1, when insulin therapy intensification due to persistent hyperglycaemia is necessary.

Reviewer 2 Report

This is a practical guideline for the detection and correction of glucocorticoid induced hyperglycemia, both in previously unknown diabetic patients, and in those with previously known diabetes. It provides updated guidelines for detection of hyperglycemia with thresholds and clock times well indicated. The respective places for oral antidiabetic drugs, and the new ones (SGL2 inhibitors ans GLP1 R agonists), and insulin are well taken. All guidelines are based on previously published recommendations (especially the British Society and the ADA).

The different settings (hospitalized patients, outpatients, and hospital discharge) are well taken.

In all, I have only one suggestion, dealing with how to handle insulin regimens in patients with type 1 diabetes, or those with type 2 diabetes already on a basal-bolus insulin regimen. A special chapter should be elaborated, with suggestions on which insulin types (basal/bolus, and corresponding doses for those on CSII) to increase/decrease, and which clock times should be considered (typically,  to be aware of a potential fall in BG at late night time).

Author Response

This is a practical guideline for the detection and correction of glucocorticoid induced hyperglycemia, both in previously unknown diabetic patients, and in those with previously known diabetes. It provides updated guidelines for detection of hyperglycemia with thresholds and clock times well indicated. The respective places for oral antidiabetic drugs, and the new ones (SGL2 inhibitors ans GLP1 R agonists), and insulin are well taken. All guidelines are based on previously published recommendations (especially the British Society and the ADA). The different settings (hospitalized patients, outpatients, and hospital discharge) are well taken.

In advance, thank you for reviewing our manuscript and your positive feedback.   

In all, I have only one suggestion, dealing with how to handle insulin regimens in patients with type 1 diabetes, or those with type 2 diabetes already on a basal-bolus insulin regimen. A special chapter should be elaborated, with suggestions on which insulin types (basal/bolus, and corresponding doses for those on CSII) to increase/decrease, and which clock times should be considered (typically,  to be aware of a potential fall in BG at late night time).

The handling of SIHG in hospitalized patients with type 1 diabetes is complex and challenging in daily routine. Hardly any evidence is available on how to adapt insulin therapy during the course of glucocorticoid therapy which can be accused to the fact that people with type 1 diabetes react very individually on exogenous steroid therapy. This limited chance of therapeutic recommendations has initially held us back to discuss this population. However, following your advice, we added a paragraph focusing on the hospitalized patient with type 1 diabetes and provide a careful and pragmatic suggestion for better understanding the glycaemic management in this depicted population of interest.  

Reviewer 3 Report

The  present paper adresses an important  topic in diabetes treatment in special situations, which is steroid induced hyperglycemia. The paper is well written and structured. However, it adds no novelty to current published reviews of the topic and several important aspects should be adressed:

Introduction: weight gain is considered as an inmediate glucocorticoid effect and hyperglycemia as gradual effect, while we know that hyperglycemia occurs in the first three days of glucocorticoid treament and weight  needs more time to increase. Authors should reconsiders this statement.

Section 6 (admission to the hospital). which is the basis of performing a lunch of 1-2h post lunch or evening glucose determination prior GC initiation in people without diabetes? The  determination of HbA1c  may  be  sufficient in this situation. if capillary glucose recommended why not fasting glucose?

Section 8.1: considering the limited use of oral antihyperglycemic agents  in the hospital setting, this the discussion offered is too long.

Scenario 1: (hydrocortisone): sulfonilurea is recommended in this setting but its  use in hospitalized patients is  not recommeneded in any guideline and its long lasting effect does  not make it an appropiate drug in this setting.

Furthermore, when discussing the  utility of rapid insulin, the authors suggest  that carbohydrate counting is useful. Considering the hospital setting, and that the use of hydrocortisone in high doses will be  implemented for a short  period of time and in patients  with acute illness, training  in carb counting may not be advisable.

Scenario 3: long-acting glucocorticoids: an initial dose of insulina glargine of 0,2 UI/Kg BW  is advised irrespective of the dose of  dexamethasone prescribed?

Figure 1:

Consider removal of the recommendation of use of SFU in the hospital setting.

Therapy adjustment:  In the case  of short acting  or intermediate acting glucocorticoids adjustment is advised according to fasting glucose when hyperglycemia occurs mainly during the day. Reconsider this recommendation.

Reconsider the recomendation of using insulin glargine in patients taking prednisolone as physiology of hyperglycemia does  not fit with glargine pharmacokinetics.

Author Response

The present paper adresses an important  topic in diabetes treatment in special situations, which is steroid induced hyperglycemia. The paper is well written and structured. However, it adds no novelty to current published reviews of the topic and several important aspects should be adressed:

We would like to thank the reviewer for the thorough review.

Introduction: weight gain is considered as an inmediate glucocorticoid effect and hyperglycemia as gradual effect, while we know that hyperglycemia occurs in the first three days of glucocorticoid treament and weight  needs more time to increase. Authors should reconsiders this statement.

 We agree with the reviewer to classify steroid induced hyperglycaemia as immediate effect of GC therapy and changed it accordingly

Section 6 (admission to the hospital). which is the basis of performing a lunch of 1-2h post lunch or evening glucose determination prior GC initiation in people without diabetes? The  determination of HbA1c  may  be  sufficient in this situation. if capillary glucose recommended why not fasting glucose?

 We appreciate the careful reading and recognized that these measurements should be done in patients who are already on steroid therapy. We have changed it accordingly.

Section 8.1: considering the limited use of oral antihyperglycemic agents  in the hospital setting, this the discussion offered is too long.

 We agree that this section contains too much details. Following your advice to shorten this section, we decided to delete the “bullet-point” part. 

Scenario 1: (hydrocortisone): sulfonilurea is recommended in this setting but its  use in hospitalized patients is  not recommeneded in any guideline and its long lasting effect does  not make it an appropiate drug in this setting.

The position statement of the JBDS (Roberts, et. al Diabetic Medicine, 2018) suggests the use of sulfonylurea in people with mild or transient hyperglycemia following steroids. However, they do not specifically associate this recommendation to hospitalized patients. For this reason we deleted it in this section.

Furthermore, when discussing the  utility of rapid insulin, the authors suggest  that carbohydrate counting is useful. Considering the hospital setting, and that the use of hydrocortisone in high doses will be  implemented for a short  period of time and in patients  with acute illness, training  in carb counting may not be advisable.

 We agree that training on carb counting in hospitalized patients experiencing (mostly) transient hyperglycaemia is not meaningful. We have deleted this sentence. 

Scenario 3: long-acting glucocorticoids: an initial dose of insulina glargine of 0,2 UI/Kg BW  is advised irrespective of the dose of  dexamethasone prescribed?

 This recommendation is based on the circumstance that relevant hyperglycaemia (defined as fasting glucose >200 mg/dl) is present. Therefore, we estimate the suggested factor as appropriate and safe. 

Figure 1:

Consider removal of the recommendation of use of SFU in the hospital setting.

SFUs were removed from the recommendation.

Therapy adjustment:  In the case  of short acting  or intermediate acting glucocorticoids adjustment is advised according to fasting glucose when hyperglycemia occurs mainly during the day. Reconsider this recommendation.

We agree that it might be the wrong measure taken to increase bolus insulin in case of persisting hyperglycaemia at the next day. More preferential might be the introduction of basal insulin in such a case. We changed this in the figure.

Reconsider the recomendation of using insulin glargine in patients taking prednisolone as physiology of hyperglycemia does  not fit with glargine pharmacokinetics.

We agree that glargine U100 is dedicated to be injected only once daily. However some kinetic data (Porcellati, Diabetes Care 2019) and daily practice indicate that iGlar U100 does not provide a complete 24 hours insulin coverage which might allow an application twice daily. But we agree that this recommendation is not sufficiently supported by research evidence and we changed the recommendation.

Round 2

Reviewer 3 Report

The  authors have adressed each of the comments raised. However, there are still some issues  that have not been solved, and some recomendations are not in accordance with the physiologic action of glucocorticoids:

For example, the  use of a unique dose of insulin glargine when using intermediate acting glucocorticoids. It is associated with poresence of nocturnal hypoglycemia, as the pharmacodynamics  on insulin glargine do not fit the hyperglycemia pattern of intermediate GC.

Another example is the  recomendation of uptitratin all doses of insulin in type 1 diabetes no matter the type of glucocorticoid is used. The changes in different insulins may be  in accordance with the pattern of the clucocorticoid and prefferentially affect rapid-acting insulin as postprandial hyperglycemia is prominent.

Section 8.1 is still very extense considering the use of oral drugs in the hospital setting. Furthermore the  reader is not able to distinguish when the author adresses type 2 diabetes in the hospital OR GC-induced hyperglycemia (tha last paragraph for example indicates the use of metformin, which would not be  indicated for acute use  of glucocorticoids due to its slow initiation of action.

Furthermore, the text should be better structured:

Section 8.2 is centered in insulin use and contain different scenarios. Scenario 1, rediscusses the  use  of oral drugs. This should be  deleted from here.

Finally the  new sections added at end of the manuscript, although are  appropiate, they should be improved. Recomendations in type 1 diabetic  patients are  only supported in one reference. However, previous reviews have already considered how to adjust treatment in insulin- dependnet diabetes taking  GC.

Author Response

The  authors have adressed each of the comments raised.

We thank this reviewer again for his careful review and his rationale amendments.

However, there are still some issues  that have not been solved, and some recomendations are not in accordance with the physiologic action of glucocorticoids:

For example, the  use of a unique dose of insulin glargine when using intermediate acting glucocorticoids. It is associated with poresence of nocturnal hypoglycemia, as the pharmacodynamics  on insulin glargine do not fit the hyperglycemia pattern of intermediate GC.

We agree that the kinetics of intermediate-acting glucocorticoids fit best to the kinetics of NPH insulin, which should be the first choice. However, there are two randomized controlled trials 10.1111/dom.12859. and 10.1016/j.diabres.2015.09.015) that confirmed the efficacy and safety of insulin glargine U100 when compared to NPH. We changed the language to make the preference clear but to also mention the available literature. In the therapy figure we weakened the recommendation for glargine U100 and mentioned it “to be considered”. 

Another example is the  recomendation of uptitratin all doses of insulin in type 1 diabetes no matter the type of glucocorticoid is used. The changes in different insulins may be  in accordance with the pattern of the clucocorticoid and prefferentially affect rapid-acting insulin as postprandial hyperglycemia is prominent. Finally the  new sections added at end of the manuscript, although are  appropiate, they should be improved. Recomendations in type 1 diabetic  patients are  only supported in one reference. However, previous reviews have already considered how to adjust treatment in insulin- dependnet diabetes taking  GC.

Guidelines or scientific reviews which explicitly focus on the behavior in insulin handling during steroid therapy in patients with type 1 diabetes are not available. The Joint British Diabetes Society provides only weak recommendations for its management (https://www.diabetes.org.uk/resources-s3/2017-09/JBDS%20management%20of%20hyperglycaemia%20and%20steriod%20therapy_0.pdf). A single mechanistic study described an additional total insulin dose requirement during 60 mg of prednisolone therapy of 30-100% (Bevier, JDST, 2008). Another study in pregnant women demonstrated a 50% fold higher need of insulin (Dashora, Diab Med, 2004). Based on this little evidence we provided an additional paragraph for recommendations of careful dose adjustments in type 1 diabetes taking the GC dose and agent into account.

Section 8.1 is still very extense considering the use of oral drugs in the hospital setting. Furthermore the  reader is not able to distinguish when the author adresses type 2 diabetes in the hospital OR GC-induced hyperglycemia (tha last paragraph for example indicates the use of metformin, which would not be  indicated for acute use  of glucocorticoids due to its slow initiation of action.

Furthermore, the text should be better structured

We adapted the paragraph, according to your suggestions indicating that some agents might be continued in depicted stable patients but not being introduced in case of acute initiation of GC therapy or acute illness.

Section 8.2 is centered in insulin use and contain different scenarios. Scenario 1, rediscusses the  use  of oral drugs. This should be  deleted from here.

We deleted the sentence focusing on oral antidiabetic drugs in this section.